# Intraoperative Extra Corporeal Membrane Oxygenator for Lung Cancer Resections Does Not Impact Circulating Tumor Cells

**DOI:** 10.3390/cancers14205004

**Published:** 2022-10-13

**Authors:** Francesco Petrella, Laura Zorzino, Samuele Frassoni, Vincenzo Bagnardi, Monica Casiraghi, Claudia Bardoni, Shehab Mohamed, Valeria Musso, Emanuele Simonini, Fabiana Rossi, Francesco Alamanni, Marco Venturino, Lorenzo Spaggiari

**Affiliations:** 1Department of Thoracic Surgery, IRCCS European Institute of Oncology, 20141 Milan, Italy; 2Department of Oncology and Hemato-Oncology, University of Milan, 20122 Milan, Italy; 3Division of Laboratory Medicine, IRCCS European Institute of Oncology, 20141 Milan, Italy; 4Department of Statistics and Quantitative Methods, University of Milano-Bicocca, 20126 Milan, Italy; 5Department of Cardiovascular Surgery, IRCCS Centro Cardiologico Monzino, 20138 Milan, Italy; 6Department of Clinical Sciences and Community Health, University of Milan, 20122 Milan, Italy; 7IRCCS Galeazzi Sant’Ambrogio, Cardiochirurgia Universitaria, 20157 Milan, Italy; 8Department of Anesthesiology, IRCCS European Institute of Oncology, 20141 Milan, Italy

**Keywords:** extracorporeal membrane oxygenator, circulating tumor cells, lung cancer, lung resection

## Abstract

**Simple Summary:**

The diagnosis of active neoplastic disease was traditionally judged an absolute contraindication for extracorporeal membrane oxygenator (ECMO) because of long-term results uncertainty and co-existing acquired coagulation disorders often diagnosed in this group of patients. There is a growing body of evidence that circulating tumor cells (CTCs) can be detected in the blood of patients before the primary tumor is diagnosed and in the case of carcinoma recurrence; moreover, on some occasions, they persist in the blood of patients after radical resection of the primary tumor. The aim of this prospective, two-arm study is to compare the number of CTCs before and after surgery in patients undergoing lung cancer resection with and without intraoperative ECMO support. Intraoperative ECMO for lung cancer resections did not impact CTC variation after the procedure and did not impact postoperative complications, ICU stay, hospital total length of stay, and post operative C-reactive protein increase.

**Abstract:**

Background: The diagnosis of active neoplastic disease was traditionally judged an absolute contraindication for extracorporeal membrane oxygenator (ECMO) because of the fear of tumor cells being scattered or seeded. The aim of this study is to compare the number of circulating tumor cells (CTCs) before and after surgery in patients receiving lung cancer resection with and without intraoperative ECMO support. Methods: This is a prospective, non-randomized, two-arms observational study comparing the number of CTCs before and after surgery in patients receiving lung cancer resection with and without intraoperative ECMO support. The ECMO arm includes patients suffering from lung cancer undergoing pulmonary resection with planned intraoperative ECMO support. The non-ECMO arm includes patients suffering from non-early-stage lung cancer undergoing pulmonary resection without planned intraoperative ECMO support. Results: Twenty patients entered the study, eight in the ECMO arm and twelve in the non-ECMO arm. We did not observe any significant difference between the ECMO and non-ECMO groups in terms of postoperative complications (*p* = 1.00), ICU stay (*p* = 0.30), hospital stay (*p* = 0.23), circulating tumor cells’ increase or decrease after surgery (*p* = 0.24), and postoperative C-reactive protein and C-reactive protein increase (*p* = 0.80). The procedures in the non-ECMO arm were significantly longer than those in the ECMO arm (*p* = 0.043). Conclusions: Intraoperative ECMO for lung cancer resections did not impact CTC increase or decrease after the procedure.

## 1. Introduction

The diagnosis of active neoplastic disease was traditionally judged an absolute contraindication for extracorporeal membrane oxygenator (ECMO) because of long-term results uncertainty and co-existing acquired coagulation disorders often diagnosed in this group of patients [1].

This attitude has slightly changed in the last decades—as disclosed by the analysis of the Extracorporeal Life Support Organization (ELSO) registry—showing a 32% survival rate at hospital discharge in adult patients with active malignancy who received extracorporeal life support (ECLS) [2]. Nevertheless, the vast majority of these patients, suffering from solid tumors, received veno-venous ECMO (VV—ECMO) as a rescue treatment for respiratory failure; on the other hand, intraoperative use of ECMO for elective oncologic thoracic surgery is still widely debated. In fact, a kind of reluctancy toward ECMO support for selected lung cancer and neoplastic airway resections is still observed in daily clinical practice, probably due to the fear of tumor cells being scattered or seeded by extracorporeal circulation [3].

There is a growing body of evidence that circulating tumor cells (CTCs) can be detected in the blood of patients before the primary tumor is diagnosed and in the case of carcinoma recurrence; moreover, on some occasions, they persist in the blood of patients after radical resection of the primary tumor [4]. In addition, the proven observation that CTCs derive from the primary tumor strengthens the hypothesis that they reflect tumor burden at different stages of neoplastic progression [5].

For all these reasons, CTCs play a crucial role not only in the early diagnosis and prognostic assessment of many solid cancers but also in defining tumor reaction to external stimuli, such as therapies and surgical treatments [6,7]. Last—but not least—they can be easily found by a simple blood test, thus replacing more aggressive approaches, such as repeated bone marrow aspiration [8].

The aim of this prospective, non-randomized, two-arm study is to compare the number of CTCs before and after surgery in patients undergoing lung cancer resection with and without intraoperative ECMO support.

## 2. Materials and Methods

### 2.1. Study Design

This is a prospective, non-randomized, two-arm observational study comparing the number of CTCs before and after surgery in patients undergoing lung cancer resection with and without intraoperative ECMO support.

The study was approved by the European Institute of Oncology Institutional Review Board (R1005/19-IEO1060—26 June 2019) and was registered at ClinicalTrials.gov (Identifier: NCT04048512).

The trial was conducted in accordance with the Declaration of Helsinki and all applicable local regulations. All the patients provided written informed consent.

### 2.2. Eligibility, Patient Population, and Management

ECMO arm: patients suffering from lung cancer undergoing pulmonary resection with planned intraoperative ECMO support.

NON-ECMO arm: patients suffering from non-early-stage lung cancer undergoing pulmonary resection without planned intraoperative ECMO support, requiring major pulmonary resection and with higher ki 67% on preoperative biopsy.

Exclusion criteria: age younger than 18 years; contraindications to general anesthesia; poor general clinical conditions (ECOG PS >= 2); patients unable to provide informed consent; metastatic patients.

### 2.3. Clinical Evaluation and Procedures

Pre-surgery: Operability was determined by standard clinical and radiographic procedures (whole-body computed tomography), nuclear imaging (whole-body fluorodeoxynucleotide positron emission tomography), and staging procedures, including endobronchial ultrasonographic bronchoscopy and transbronchial needle aspiration, as appropriate. Preoperative respiratory function was assessed routinely by blood gas analysis and spirometry; lung perfusion scanning to identify the functionally prevalent lung and cardiopulmonary exercise test were performed in cases of planned pneumonectomy.

In case of anatomical lung resection, the pulmonary vein was ligated and transected first whenever possible.

Post-surgery: Amoxicillin and clavulanic acid were administered for the first five postoperative days (or less in the case of shorter admissions) in nonallergic patients, with the first dose administered before the skin incision. Thromboprophylaxis was maintained with sequential compression devices, early ambulation, and low molecular-weight heparin delivered subcutaneously (4000 IU/d enoxaparin sodium). Postoperative analgesia was achieved in conjunction with anesthesiologists using a combination of epidural analgesia (when technically feasible and not contraindicated), patient-controlled analgesia, and oral and parenteral adjuncts as needed to improve pulmonary and physical therapy.

ECMO management: Veno-venous ECMO was set up by percutaneous cannulation of the right femoral and jugular veins (CardioHelp System, Maquet, Getinge Group, Rastatt, Germany); activated clotting time (ACT) was recorded for each patient before and after systemic low-heparinization. ECMO flow, O_2_% saturation, and arterial pressure were recorded at times of 0, 10, 20, and 30 min after ECMO start. Ventilation arrest and total ECMO duration were also recorded. Extracorporeal circulation was strictly limited to the phase of the procedure requiring ventilation arrest, thus minimizing its duration and the required anticoagulation.

Postoperative Complications: Postoperative death was defined as 30-day and 90-day mortality or longer if mortality occurred during hospitalization. Complications were classified according to the thoracic morbidity and mortality classification system [9] and the Clavien–Dindo classification.

In-Hospital and ICU Length of Stay: In-hospital length of stay was defined as the time spent in the hospital from the day of operation (operative day 0) to discharge. ICU length of stay was defined as the time spent in the ICU from the day of admission to discharge to the ward.

### 2.4. Circulating Tumor Cells Assessment

Peripheral blood samples were collected in dedicated tubes (CellSave preservative tubes). Twenty milliliters of blood were taken at each time of sampling (24 h before surgery and the morning of post operative day 1), and samples were kept at room temperature and processed within 96 h of collection.

The quantification and direct isolation of circulating tumor cells was performed by the CellSearch System (Menarini Silicon Biosystems, Bologna, Italy). The method has been described in detail elsewhere [4]. Briefly, ferrofluid particles conjugated to anti-EpCAM antibodies are used for the isolation of EpCAM-positive cells by means of a magnetic field. After removing the supernatant, the cells are fluorescently stained using 4’,6-diamidino-2-phenylindole (DAPI) for nucleic acid, anti-cytokeratin-phycoerythrin for epithelial cells, and allophycocyanin-conjugated anti-CD45 antibody to detect and exclude leucocytes. Stained cells are then analyzed with a fluorescence microscope that automatically scans the surface of the reaction cartridge, and CTCs are defined as nucleated cells (DAPI positive) lacking CD45 but expressing cytokeratins (CKs). The presence of one or more cells per 7.5 mL of blood in any of the two tubes collected at each sampling time was considered to be a positive result (Figure 1).

Example of a fluorescent microscopy CTC analysis: Reading from left to right in the same row, the first colored image is a falsely colored composite image of the cell cytoplasm (green) and nucleus (magenta), the second is the image of the cytokeratin-stained cytoplasm (CK-PE), the third is the image of a DAPI-stained nucleus, the fourth is the gray-scale image of the CD45-APC, and the fifth is used for instrument calibration and additional phenotyping. The events n° 10-12-13 show an image scored as a CTC. A cell is classified as a CTC based on its morphologic features (nearly round or oval morphology with a visible nucleus within the cytoplasm) and staining patterns (DAPI positive, cytokeratin positive, and CD45 negative). CK-PE, phycoerythrin-conjugated antibodies that bind to cytokeratins 8, 18, and 19; DAPI, nuclear dye 4,6-diamidino-2-phenylindole; CD45-APC, monoclonal antibody specific for leukocytes conjugated to allophycocyanin.

### 2.5. Outcome Measures

The primary endpoint was the increase or decrease of CTCs after the procedure compared to the number of CTCs before the procedure. Secondary outcomes of interest included duration of surgical procedure, postoperative CRP increase compared to preoperative CRP (measured at the same time of CTCs), post operative complications, intensive care unit (ICU), and hospital length of stay (LOS). 

### 2.6. Statistical Analysis

Continuous data were reported as median and range, while categorical data were reported as counts and percentages.

Wilcoxon’s signed-rank test for continuous variables and Fisher’s exact test for categorical variables were used to compare the distribution of the patients’ demographics and the clinical and outcome variables between the two groups (ECMO vs. non-ECMO).

All analyses were performed using SAS software v.9.4 (SAS Institute, Cary, NC, USA). A *p*-value less than 0.05 was considered statistically significant.

## 3. Results

Twenty patients entered the study, eight in the ECMO arm and twelve in the non-ECMO arm. Median age was 68 (38–75) in the ECMO arm and 65 (53–79) in the non-ECMO arm; there were six male patients (75%) in the ECMO arm and nine (75%) in the non-ECMO arm. Median body mass index (BMI) was 24.0 (18.0–34.9) in the ECMO arm and 25.3 (17.7–34.1) in the non-ECMO arm. Five patients (63%) were operated on the right side in the ECMO arm, and six patients (50%) were operated on the right side in the non-ECMO arm. In the ECMO group, the final pathology disclosed was adenocarcinoma in five patients (62.5%); cystic adenoid carcinoma in one patient (12.5%); squamous cell carcinoma in one patient (12.5%); myofibroblastic inflammatory tumor in one patient (12.5%). In the non-ECMO group, the final pathology disclosed was: adenocarcinoma in five patients (42%); squamous cell carcinoma in six patients (50%); poorly differentiated carcinoma in one patient (8%).

Histological diagnosis was confirmed in all cases prior to surgery in 17 cases by bronchoscopy/EBUS TBNA and in 3 cases by transthoracic CT-guided biopsy.

In the ECMO group four patients (50%) underwent induction treatments, while in the non-ECMO group five patients (42%) received induction treatments. In the ECMO group two patients (25%) received adjuvant treatment, while in the non-ECMO group five patients (42%) underwent adjuvant treatment (Table 1).

Two cases of right tracheal sleeve pneumonectomy—with planned intraoperative ECMO—were shifted to standard right pneumonectomy without carinal reconstruction, thus not requiring ECMO support; both patients were therefore shifted to the control group.

Procedures performed in the ECMO arm were: right tracheal sleeve pneumonectomy in three cases, wedge resection on postpneumonectomy single lung patient in four cases (two robot-assisted, one video assisted, and one open), right upper lobectomy in one patient with previous left lobe resection and limited pulmonary function.

Procedures performed in the non-ECMO arm were: five pneumonectomies, five lobectomies, one wedge resection, and one bronchial fistula closure by Abruzzini technique.

Median ECMO duration was 60 min (range: 39–132); median ventilation arrest was 46 min (range 20–124); median preoperative ACT was 143 s (range: 108–157); median preoperative ACT after systemic heparinization was 275 s (246–355). Mean ECMO flows at T0, T10, T20, and T30 were, respectively, 963 mL (range: 200–2000), 2213 mL (range: 2000–2700), 2575 mL (range: 2260–3000), and 2625 mL (2170–3080). Mean O_2_% saturations at T0, T1, T2, and T3 were, respectively, 100% (97 -100), 98% (range 95–100), 97% (range 95–100), and 96% (range 95–99). Mean arterial pressure values (systolic/diastolic) were at T0, T1, T2, and T3, respectively, T0 87/56 (83–113/51–76); T1 96/58(90–152/51–77); T2 127/72 (109–137/59–88); and T3 123/60 (70–140/45–91) (Table 2).

None of the patients in the ECMO group had post-op CTCs greater than pre-op CTCs, while this was the case in three patients of the non-ECMO group (25%, *p* = 0.24). Details of pre- and post-op CTC distribution are reported in Figure 2. 

The median differences between post- and pre-op CRP were 8.7 (range: −0.4–17.4) in the ECMO group and 9.0 (range: 2.5–20.5) in the non-ECMO group (*p* = 0.80). The procedures related to the non-ECMO patients were significantly longer (with a median of 176 min) than those related to the ECMO ones (with a median of 105 min, *p* = 0.043). There were four patients with post-op complications in the ECMO group (50%) (two had grade 2 complications according the Clavien–Dindo classification, one had grade IIIb, and one had grade IV) and six in the non-ECMO group (five patients had grade II complications according to the Clavien–Dindo classification and one hade grade IIIb) (50%, *p* = 1.00).

30-day and 90-day mortalities were 0% in both groups.

The median ICU stay was 1 day in both the ECMO and non-ECMO patients (*p* = 0.30), while the median hospital stay was 15 days (range: 3–25) in the ECMO group and 7 days (range: 4–10) in the non-ECMO patients (*p* = 0.23). The status at the last follow-up (with a median follow-up of 6.1 months) was similar between the two groups, with one death (13%) and two patients alive with disease (25%) in the ECMO group and two deaths (17%) and two patients alive with disease (17%) in the non-ECMO group (Table 3).

## 4. Discussion

During the last ten years, the use of ECMO for thoracic tumors has increased more than 14 fold [10]. This is probably due the global development of ECMO centers, resulting in easier access [11]. On the contrary, the ratio of ECMO use for chest tumors to all adult ECMO runs has not changed during the last twenty years, thus implying an increase in total ECMO use rather than a modification in the threshold for application with this group of patients [10]. This attitude still reflects the reluctancy of many groups toward the use of ECMO in oncologic populations because of the fear of tumor cell seeding or scattering by extracorporeal circulation. Lung cancer represents about half of thoracic cancers treated by ECMO; this is due to its high prevalence—lung cancer being the second most common cancer and the leading cause of cancer-related death in the world—and to the fact that it directly affects respiration in a multitude of ways [12].

While the use of ECMO in the context of acute respiratory failure after pneumonectomy is not encouraged because of very poor prognosis [13], its application in the elective intraoperative scenario for pneumonectomy needing tracheobronchial reconstruction or upper airway extended resections is becoming an interesting alternative to cross-field ventilation [14]. 

In the recent experience of Suzuky et al.—from the Extracorporeal Life Support Organization (ELSO) Registry Analysis—upper airway neoplasm was the only oncologic group disclosing a better prognosis than all adult ECMO runs (73.5%); similarly, the use of ECMO for tracheobronchial procedures showed a favorable prognosis (66.7%) [10]. This is probably due to ECMO indications in these populations, which are most related to airway local impairment rather than to long-term functional lung disfunction. In our experience, ECMO support was started just before performing the part of the procedure needing respiratory assistance and was stopped at the end after careful checking standard orotracheal ventilation efficacy. In this way we limited both ECMO duration and flow, thus minimizing the need for anticoagulation. As a consequence, we did not observe a higher complication rate in the ECMO group; in particular, postoperative bleeding and postoperative CRP values were almost identical in the two groups, thus suggesting a similar inflammatory postoperative status without any impact on ECMO support.

For low volume ECMO support, the maximum median flow was 2625 mL 30 min after starting the ECMO run-allowed excellent tissue perfusion, resulting in a median O_2_% saturation which never fell below 96%. Surprisingly, ECMO-assisted procedures were faster than those without ECMO support; this is probably because four out of eight ECMO-assisted procedures were wedge resections on single-lung patients, thus favorably impacting on mean total duration. In any case, although ECMO support might have helped to speed up some procedures—for example tracheobronchial reconstruction—it should be correctly emphasized that ECMO support requires additional time not only for cannulation but also for its gradual starting and weaning during the operation. Although in our experience we did not observe a higher post operative complication rate in the ECMO group or specific ECMO-related complications, it should always be taken into consideration that complications might occur during and after ECMO runs [15]. They can be divided into mechanical-related and patient-related complications; mechanical–related complications are mainly represented by oxygenator failure, circulation pipeline rupture, technical problems of the pressure sensors or the pump and, more frequently, cannula dislocation [16,17]. The incidence of this type of complication is very low in experienced ECMO centers and can be effectively prevented by careful preoperative checking of the system. On the other hand, patient–related complications, such as hemolysis, intra and postoperative bleeding, liver, heart, or kidney failure, as well as nervous system complications, metabolic disorders, and infections, are more strictly related to clinical conditions and could be minimized by proper patient selection [18,19].

There is a growing interest in modern clinical oncology regarding the isolation and characterization of CTCs because of their intrinsic properties In fact, they can be detected in the peripheral blood long before clinical detection of the primary tumor which they belong to; they are found in a considerable number of patients diagnosed with cancer recurrence; they continue to be found in some patients after primary tumor radical resection [20,21,22]. As CTCs derive from clones of the primary tumor, they might reproduce the tumor burden at all stages of cancer progression [22]; for this reason, they may not only play a crucial role in early diagnosis and prognostic evaluation but could further characterize both immunophenotypic and genetic modifications occurring with tumor progression, thus helping to better assess targeted therapy [23]. Moreover, being detected by an easy blood test on peripheral blood, CTCs can be safely and frequently evaluated for an updated oncologic overview of the patient without the need for repeated invasive procedures, such as bone barrow aspiration, which negatively affect patient compliance. It has been previously demonstrated that the CellSearch system offers a careful and reproducible analysis, even in the case of small numbers of cells and a wide morphologic variety [10]; in fact, the total number of blood epithelial cells in subjects without any clinical evidence of tumors is extremely low, thus suggesting that CTC quantification may have a significant clinical benefit in all epithelial neoplasms [10]. Nevertheless, the clinical use of this system has been limited until now mainly to breast and prostate cancers [23,24], while very few data are available on the role of CTCs in pulmonary tumors, in particular in the non-metastatic stage.

Only 20 patients were enrolled in this pilot study and among them, 8 in the ECMO arm; given the peculiarity of ECMO support for thoracic malignancies resection—at present—suitable cases are quite rare; in two cases of planned ECMO resection (right tracheal sleeve pneumonectomy), ECMO was not necessary (standard right pneumonectomy without carinal reconstruction was finally performed), and these patients were therefore shifted to the non-ECMO ARM. Although prospective, this is a non-randomized trial and, due to the intrinsic nature of the study, randomization cannot be applied for both technical and ethical reasons. Due to the different settings of ECMO-assisted resections, the histology of treated patients was quite heterogenous so we did not focus on oncologic outcomes which are not the topic of this paper. In any case, further studies are required to better assess the prognostic role of CTCs in lung cancer patients, regardless of intraoperative ECMO assistance.

## 5. Conclusions

Intraoperative ECMO for lung cancer resections did not impact CTC increase or decrease after the procedure. Moreover, it did not impact postoperative complications, ICU stay, hospital total length of stay, and post operative C-reactive protein increase.

ECMO can be therefore considered a safe and effective additional tool in selected cases of lung tumor resection without any oncologic additional impact. Further studies with longer follow ups are required to better assess the impact on oncologic prognosis of postoperative CTCs and CTC increase or decrease after surgery.

## Figures and Tables

**Figure 1 cancers-14-05004-f001:**
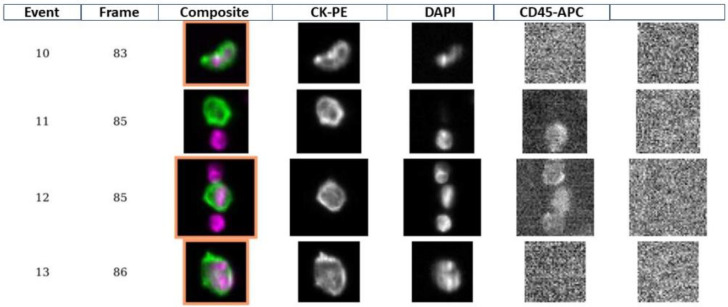
A gallery of images screened for CTCs.

**Figure 2 cancers-14-05004-f002:**
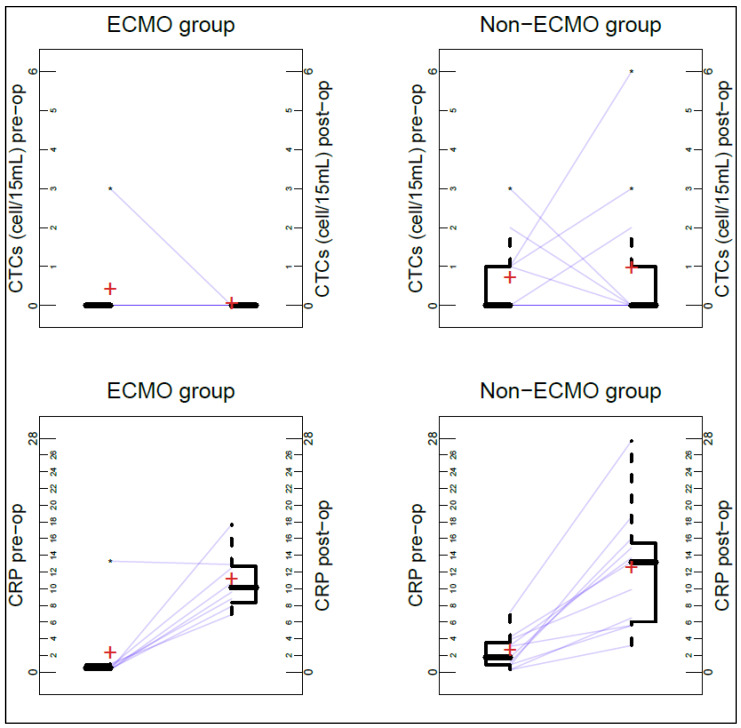
Distribution and individual trajectories of CTCs and CRP pre- and post-op, among patients with and without intraoperative ECMO. Matched box plots showing quantitative changes (pre-op and post-op) of CTCs and CRP in each patient. Half boxes represent the interquartile range and the horizontal bold lines across the boxes indicates the median. Arithmetic means are indicated with the + symbol. The asterisks indicate the values of any data points which lie beyond the extremes of the whiskers (the whiskers extend to the most extreme data point which is no more than 1.5 times the IQR away from the box).

**Table 1 cancers-14-05004-t001:** Distribution of demographic and clinical variables among patients with and without intraoperative ECMO (N = 20).

Variable	Level	ECMO (N = 8)	Non-ECMO (N = 12)	*p*-Value
Age (years), median (min–max)		68 (38–75)	65 (53–79)	1.00
Sex, N (%)	Female	2 (25)	3 (25)	1.00
	Male	6 (75)	9 (75)	
BMI (kg/m^2^), median (min–max)		24.0 (18.0–34.9)	25.3 (17.7–34.1)	0.44
ASA, N (%)	2	4 (50)	5 (45)	1.00
	3	4 (50)	6 (55)	
Tube, N (%)	Magill	5 (63)	0 (0)	0.004
	Robert Shaw Right	0 (0)	4 (36)	
	Robert Shaw Left	3 (38)	7 (64)	
Tube caliber, median (min–max)		8 (7.5–39)	39 (37–41)	0.003
Side, N (%)	Right	5 (63)	6 (50)	0.67
	Left	3 (38)	6 (50)	
Histology, N (%)	Adenocarcinoma	5 (63)	5 (42)	0.19
	Cystic adenoid carcinoma	1 (13)	0 (0)	
	Poorly differentiated carcinoma	0 (0)	1 (8)	
	Squamous cell carcinoma	1 (13)	6 (50)	
	Myofibroblastic inflammatory tumor	1 (13)	0 (0)	
TNM, N (%) ^a^	T1N0	0 (0)	1 (9.1)	-
	T2N0	1 (33)	2 (18.2)	
	T2aN1M1 (single)	1 (33)	0 (0.0)	
	T3N2	0 (0)	4 (36.4)	
	T4N0	1 (33)	2 (18.2)	
	T4N1	0 (0)	2 (18.2)	
Stage, N (%) ^b^	IB	1 (33)	3 (27)	-
	IIIA	1 (33)	4 (36)	
	IIIB	0 (0)	4 (36)	
	IVA	1 (33)	0 (0)	
Induction treatment, N (%)	No	4 (50)	7 (58)	1.00
	Yes	4 (50)	5 (42)	
Post-op treatment, N (%)	No	6 (75)	7 (58)	0.64
	Yes	2 (25)	5 (42)	

^a^ 5 Not Applicable in “ECMO” group and ^b^ not applicable in “non-ECMO” group: the statistical test was not performed.

**Table 2 cancers-14-05004-t002:** Distribution of ECMO variables among patients with intraoperative ECMO (N = 8).

Variable	Median (Min–Max) N = 8
ECMO duration (minutes)	60 (39–132)
Ventilation arrest (minutes)	46 (20–124)
Preoperative ACT (seconds)	143 (108–157)
Preoperative ACT after heparinization (seconds)	275 (246–355)
ECMO flow (mL)	
T_0_	963 (200–2000)
T_10_	2213 (2000–2700)
T_20_	2575 (2260–3000)
T_30_	2625 (2170–3080)
O_2_ saturation (%)	
T_0_	100 (97–100)
T_10_	98 (95–100)
T_20_	97 (95–100)
T_30_	96 (95–99)
Arterial pressure (diastolic, mmHg)	
T_0_	56 (51–76)
T_10_	58 (51–77)
T_20_	72 (59–88)
T_30_	60 (45–91)
Arterial pressure (systolic, mmHg)	
T_0_	87 (83–113)
T_10_	96 (90–152)
T_20_	127 (109–137)
T_30_	123 (70–140)

**Table 3 cancers-14-05004-t003:** Distribution of outcome variables among patients with and without intraoperative ECMO (N = 20).

Variable	Level	ECMO (N = 8)	Non-ECMO (N = 12)	*p*-Value
Operative time (min), median (min–max)		105 (32–242)	176 (97–263)	0.043
Post-op complications, N (%)	No	4 (50)	6 (50)	1.00
	Yes	4 (50)	6 (50)	
ICU stay (days), median (min–max)		1 (1–4)	1 (0–3)	0.30
Hospital stay (days), median (min–max)		15 (3–25)	7 (4–10)	0.23
CTCs post-op > CTCs pre-op, N (%)	No	8 (100)	9 (75)	0.24
	Yes	0 (0)	3 (25)	
CRP post-op–CRP pre-op, median (min–max)		8.7 (−0.4–17.4)	9.0 (2.5–20.5)	0.80
Status at last FU, N (%)	NED	5 (63)	8 (67)	1.00
	AWD	2 (25)	2 (17)	
	DOD	1 (13)	2 (17)	

NED: non evident disease; AWD: alive with disease; DOD: dead of disease.

## Data Availability

The data can be shared up on request.

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
