# Peer review of "Intraoperative Extra Corporeal Membrane Oxygenator for Lung Cancer Resections Does Not Impact Circulating Tumor Cells"

_cancers, 2022, doi:10.3390/cancers14205004_

Round 1

Reviewer 1 Report

Dear Authors 

Thank you for the opportunity to analyze your article.        

In this article, authors have compared the number of circulating tumor cells (CTCs) before and after surgery in patients undergoing lung cancer resection with and without intraoperative ECMO support.

            This is a well written and interesting article about a burden subject of the impact of ECMO for elective oncologic thoracic surgery on CTCs. As reported by authors and in the literature, CTCs can be identified before the primary tumor is diagnosed, and are still present after the resection of the tumor and its rate is impacted by oncologic treatments. Nevertheless, the role and impact in carcinogenesis of these cells is not yet well established. We don’t know if all CTCs are able to invade metastasis sites for example – “homing”. 

            Concerning the methodology:

            Choice of CTCs: Easy test to conduct. Deriving from the tumor, CTCs “reflects tumor burden at different stages of the tumor progression and regression, but we don’t know if all CTCs are able to invade metastasis sites for example – “homing” and CTCs rate is not completely in correlation with disease free and long-term survival. 

            Methology: In accordance with the study. Patients are their own control, surgical resections are different so, comparing ECMO group and non-ECMO group about post operative event is difficult according me. 

            Surgery: How were performed the different resections? Was the vein tied in 1st systematically for example? Or the artery? 

            Clinical Evaluation: Is it possible to study 90-d mortality rather than 30-d? 

            Post operative complication: Is it possible to report it with more details and according the Clavien Dindo classification? Do you think that complications due to ECMO can influence CTCs rate during post operative period?

            CTCs assessment: 

            -       Definition of pre and post operative? What does mean “before and after surgery” – lines 135 and 136 - Just before skin incision? Just after skin closure? Earlier? Later? Could you give more details? 

           -       According you, CTCs rate later during post operative period is interesting? CTCs rate could be modified by any post operative events, not only the ECMO, it could be confusing but ECMO lead to many “blood changes” and phsyiopathological changes latter during post operative period. 

            Outcomes: 

            -       When was measured the CRP in pre and post operative? At the same period of the measure of CTCs ?

         -       Outcomes in accordance the study, but it could be interesting to pursue a long-term follow-up, until recurrences for example, for an other article for example. 

            Concerning the results:

            Results are clearly reported except:

         -        The report of post operative complications which are not mentioned and it could be interesting to report it using the Clavien Dindo Classification.

         -       Definition of pre and post operative for CTCs assessment and CRP.

         -       Concerning the follow up could you give some details about deaths’ reason and diseases? 

            Concerning the discussion:

Well written and interesting. As you wrote, there is a “fear of tumor cell seeding or scattering by extracorporeal circulation” – lines 246 and 247 – but is it only the CTCs rate that can be related to the risk of dissemination? If CTC is unable to “homing” maybe not, but a higher rate lead to a higher risk of having one CTC that can be seed… 

Study limitations are well described and conclusions are findings are moderated about the time of surgery during an ECMO assisted resection, and the real impact of CTCs. 

As you reported, many parameters must be taken into account in order to understand the impact of ECMO, the impact of variation of CTCs on long term survival and recurrences. 

            It’s a well written, easy reading and interesting article, that need some precisions. 

            Congratulations to authors for this work. 

Author Response

Dear reviewer 1, thank you so much for your review and valuable comments and suggestions.

We did our best to answer all of your queries and to edit the revised version of the manuscript according your suggestions and considerations.

Kind regards,

REVIEWER 1

Dear Authors 

Thank you for the opportunity to analyze your article.        

In this article, authors have compared the number of circulating tumor cells (CTCs) before and after surgery in patients undergoing lung cancer resection with and without intraoperative ECMO support.

            This is a well written and interesting article about a burden subject of the impact of ECMO for elective oncologic thoracic surgery on CTCs. As reported by authors and in the literature, CTCs can be identified before the primary tumor is diagnosed, and are still present after the resection of the tumor and its rate is impacted by oncologic treatments. Nevertheless, the role and impact in carcinogenesis of these cells is not yet well established. We don’t know if all CTCs are able to invade metastasis sites for example – “homing”. 

            Concerning the methodology:

            Choice of CTCs: Easy test to conduct. Deriving from the tumor, CTCs “reflects tumor burden at different stages of the tumor progression and regression, but we don’t know if all CTCs are able to invade metastasis sites for example – “homing” and CTCs rate is not completely in correlation with disease free and long-term survival. 

            Methology: In accordance with the study. Patients are their own control, surgical resections are different so, comparing ECMO group and non-ECMO group about post operative event is difficult according me. 

  • Surgery: How were performed the different resections? Was the vein tied in 1st systematically for example? Or the artery?

In case of anatomical lung resection, the pulmonary vein was ligated and transected first whenever possible.

(Revised version, lines 110,111).

  • Clinical Evaluation: Is it possible to study 90-d mortality rather than 30-d?

We added data about mortality: 30-day and 90-dat mortality were 0% in both groups; Revised version line 241.

  • Post operative complication: Is it possible to report it with more details and according the Clavien Dindo classification? Do you think that complications due to ECMO can influence CTCs rate during post operative period?

We added the description of complications according to the Clavien – Dindo classification

There were 4 patients with post-op complications in the ECMO group (50%) (two  had grade 2 complications according the Clavien – Dindo classification, one had grade IIIb and one had grade IV)  and 6 in the non-ECMO group  (five patients had grade II complications according to the Clavien – Dindo Classification and one hade grade IIIb) (50%, P=1.00).

Revised version lines 237 – 241(Results)  and line 131, 132 (methods).

As far as we know now,  at this point of the study,  we do not think that complications due to ECMO can influence CTCs rate during post operative period

            CTCs assessment: 

  • Definition of pre and post operative? What does mean “before and after surgery” – lines 135 and 136 - Just before skin incision? Just after skin closure? Earlier? Later? Could you give more details? 

Twenty milliliters of blood were taken at each time of sampling (24 hours before surgery and the morning of post operative day 1)… (Revised version lines 137,138).

  • According you, CTCs rate later during post operative period is interesting? CTCs rate could be modified by any post operative events, not only the ECMO, it could be confusing but ECMO lead to many “blood changes” and phsyiopathological changes latter during post operative period.

To our present knowledge of the topic, CTC modification would have expected soon after the procedure; this is why we decided to test post operative  CTCs on post op day 1; anyway this is simply an hypothesis because no celar data exist nowadays on this topic.

            Outcomes: 

  • When was measured the CRP in pre and post operative? At the same period of the measure of CTCs ?

C- reactive protein was measured at the same period of the measure of the CTCs (24 hours before surgery and the morning of post operative day 1) Revised version  line 169.

  • Outcomes in accordance the study, but it could be interesting to pursue a long-term follow-up, until recurrences for example, for an other article for example

We modified  – in the conclusion section- the phrase “Further studies with longer follow up  are required to better assess the impact on oncologic prognosis of postoperative CTCs and CTC increase or decrease after surgery” to better focus on your suggestion (Revised version lines 333 – 335)

            Concerning the results:

            Results are clearly reported except:

  •  The report of post operative complications which are not mentioned and it could be interesting to report it using the Clavien Dindo Classification. 

We added the description of complications according to the Clavien – Dindo classification

There were 4 patients with post-op complications in the ECMO group (50%) (two  had grade 2 complications according the Clavien – Dindo classification, one had grade IIIb and one had grade IV)  and 6 in the non-ECMO group  (five patients had grade II complications according to the Clavien – Dindo Classification and one hade grade IIIb) (50%, P=1.00).

Revised version lines 237 – 241(Results)  and line 131, 132 (methods).

  • Definition of pre and post operative for CTCs assessment and CRP.

CTS measuring: Twenty milliliters of blood were taken at each time of sampling (24 hours before surgery and the morning of post operative day 1)… (Revised version lines 137,138).

C- reactive protein was measured at the same period of the measure of the CTCs (24 hours before surgery and the morning of post operative day 1) Revised version  line 169.

  • Concerning the follow up could you give some details about deaths’ reason and diseases? 

One patient died of disease in the ECMO arm while 2 patients died of disease in the non ECMO arm; during the follow up we did not observe any death du to other causes than tumor.

Revised table 3 , revised version line 245

            Concerning the discussion:

  • Well written and interesting. As you wrote, there is a “fear of tumor cell seeding or scattering by extracorporeal circulation” – lines 246 and 247 – but is it only the CTCs rate that can be related to the risk of dissemination? If CTC is unable to “homing” maybe not, but a higher rate lead to a higher risk of having one CTC that can be seed… 

I fully agree with your suggestion; maybe the higher the number of post op CTS, the higher the possibility of an “effective” homing; this is why we decided to quantify post op ctcs number variation, as an indirect evaluation of homing’s probability

Study limitations are well described and conclusions are findings are moderated about the time of surgery during an ECMO assisted resection, and the real impact of CTCs. 

As you reported, many parameters must be taken into account in order to understand the impact of ECMO, the impact of variation of CTCs on long term survival and recurrences. 

            It’s a well written, easy reading and interesting article, that need some precisions. 

            Congratulations to authors for this work. 

Reviewer 2 Report

Dear Editor and Authors,

It was my pleasure to review this work titled “Intraoperative Extra Corporeal Membrane Oxygenator for Lung Cancer Resections Does not Impact on Circulating Tumor Cells” by Dr. Petrella and the Dr. Spaggiari’s research team at the European Institute of Oncology in Milan, Italy.

I must say that just from the title of the manuscript my interest was peaked!! For years we have thought that Cardiopulmonary Bypass (CPB) and Extra Corporeal Membrane Oxygenation (ECMO) are contraindicated in the presence of concurrent, active cancer because there is a risk of dissemination of disease. In this prospective, non-randomized, two arm single institution study the authors attempt to investigate if the above premise could be true by investigating the amount of blood circulating tumor cells before and after a lung resection with the use of ECMO and without. The authors have studied blood samples from 20 patients assigned into two groups; 8 in the ECMO arm and 12 in the non-ECMO one. This analysis demonstrated that there is no significant increase or decrease in the number of circulating tumour cells in either group.

This is as I previously alluded to, an interesting concept and study with significant clinical interest. It is well conducted and reported, in succinct and understandable language with very minor syntax or expression errors.  The manuscript is well structured and illustrated with instructive and clear tables and flows well to the reader.

There are however some issues I would like addressed by the authors prior to my final assesment:

1. In terms of the study’s inclusion criteria, it is clearly stated how the ECMO group was selected (planned intraporative ECMO use) however we are not told how the non-ECMO group was selected. Were the patients analyzed consecutive patients?

2. Was histological diagnosis confirmed in all cases prior to surgery? How was this achieved? Bronchoscopy/EBUS or CT-FNB.

3. In terms of circulating tumor cell assessment the timing of sampling is not clearly mentioned (was it done at the ward, in the OR prior to anesthesia induction ect).

4. It is interesting that the groups contain one patient of stage IB in the ECMO arm and four patients stage IB in the non-ECMO arm. This seems to contradict the stated inclusion criteria “NON-ECMO arm: patients suffering from non-early-stage lung cancer undergoing 97 pulmonary resection without planned intraoperative ECMO support”. Can the authors comment on this? Why was ECMO planned/used for a stage IB resection?

5. In table 3, NED, AWD and DED need to be defined please.

6. Were the wedge resections in the post-pneumonectomy/single lung patients performed as an open procedure or VATS?

7. Why was a bronchial fistula closure by Abruzzini technique included in the analysis as it is not a lung resection? Indeed staging is missing for one patient in the non-ECMO arm were only 11 patients are reported! Why would CTC be present in this patient and why analyze for them if this patient had his lung resection at an earlier stage and presumably developed a BPF subsequently. This, patient does not fit the inclusion criteria and possibly needs to be removed from the analysis!  

8. If the above patient is kept in the analysis the authors need to edit lines 321-322 “we enrolled only lung cancer patients undergoing pulmonary resection and excluded patients receiving ECMO support for other surgical indications, such as mediastinal and thymic resections.”

In conclusion, as previously mentioned this is an interesting study with clinical applicability in surgical practice. As such I am keen for it to be presented to the surgical community. However, there are some significant methodological and study conduction issues that need addressing prior to doing so.  I wish well to all.

My kindest regards,

Author Response

Dear reviewer 2, thank you so much for your review and valuable comments and suggestions.

We did our best to answer all of your queries and to edit the revised version of the manuscript according your suggestions and considerations.

Kind regards,

Dear Editor and Authors,

It was my pleasure to review this work titled “Intraoperative Extra Corporeal Membrane Oxygenator for Lung Cancer Resections Does not Impact on Circulating Tumor Cells” by Dr. Petrella and the Dr. Spaggiari’s research team at the European Institute of Oncology in Milan, Italy.

I must say that just from the title of the manuscript my interest was peaked!! For years we have thought that Cardiopulmonary Bypass (CPB) and Extra Corporeal Membrane Oxygenation (ECMO) are contraindicated in the presence of concurrent, active cancer because there is a risk of dissemination of disease. In this prospective, non-randomized, two arm single institution study the authors attempt to investigate if the above premise could be true by investigating the amount of blood circulating tumor cells before and after a lung resection with the use of ECMO and without. The authors have studied blood samples from 20 patients assigned into two groups; 8 in the ECMO arm and 12 in the non-ECMO one. This analysis demonstrated that there is no significant increase or decrease in the number of circulating tumour cells in either group.

This is as I previously alluded to, an interesting concept and study with significant clinical interest. It is well conducted and reported, in succinct and understandable language with very minor syntax or expression errors.  The manuscript is well structured and illustrated with instructive and clear tables and flows well to the reader.

There are however some issues I would like addressed by the authors prior to my final assesment:

  1. In terms of the study’s inclusion criteria, it is clearly stated how the ECMO group was selected (planned intraporative ECMO use) however we are not told how the non-ECMO group was selected. Were the patients analyzed consecutive patients?

We enrolled in the non-ECMO arm those patients with higher probability to have CTC before the resection (advanced stage, N2 disease, planned pneumonectomy); we did not use consecutive enrollment criteria

Revise version lines 97 - 99

  1. Was histological diagnosis confirmed in all cases prior to surgery? How was this achieved? Bronchoscopy/EBUS or CT-FNB.

Histological diagnosis was confirmed in all cases prior to surgery, in 17 cases by Bronchoscopy/EBUS TBNA, in 3 cases by transthoracic CT-guided biopsy (Revised version lines 192,193)

  1. In terms of circulating tumor cell assessment the timing of sampling is not clearly mentioned (was it done at the ward, in the OR prior to anesthesia induction ect).

Twenty milliliters of blood were taken at each time of sampling (24 hours before surgery and the morning of post operative day 1)… (Revised version lines 137,138).

  1. It is interesting that the groups contain one patient of stage IB in the ECMO arm and four patients stage IB in the non-ECMO arm. This seems to contradict the stated inclusion criteria “NON-ECMO arm: patients suffering from non-early-stage lung cancer undergoing 97 pulmonary resection without planned intraoperative ECMO support”. Can the authors comment on this? Why was ECMO planned/used for a stage IB resection?

The stage Ib patient in the ECMO arm was the patient undergoing right upper lobectomy  with previous left lobe resection and limited pulmonary function (Revised version lines 210 – 211).

With regard to 3 patients in the non-ECMO arm they were enrolled because of higher Ki 67 values diagnosed on preoperative biopsy, thus suggesting an aggressive disease with higher chance of detectable CTCs. One of them was submitted to pneumonectomy because of very centrally located lesion. (revised version line 99)

  1. In table 3, NED, AWD and DED need to be defined please.

At the bottom of table 3 we added the description : NED: non  evident disease; AWD: alive with disease; DOD: dead of disease.

  1. Were the wedge resections in the post-pneumonectomy/single lung patients performed as an open procedure or VATS?

wedge resection on postpneumonectomy single lung patient in 4 cases (2 robot-assisted, 1 video assisted and 1 open) Revised version lines 204,205

  1. Why was a bronchial fistula closure by Abruzzini technique included in the analysis as it is not a lung resection? Indeed staging is missing for one patient in the non-ECMO arm were only 11 patients are reported! Why would CTC be present in this patient and why analyze for them if this patient had his lung resection at an earlier stage and presumably developed a BPF subsequently. This, patient does not fit the inclusion criteria and possibly needs to be removed from the analysis!  

The Abruzzini case was included because the procedure was realized soon after a pneumonectomy of a patient with higher ki 67%, thus suggesting an aggressive local disease with higher chance to find CTC in the blood.

The patient in the non-ECMO group that was not staged was the one receiving a wedge resection on contralateral lung after contralateral lobectomy, without lymphadenectomy, thus complete staging was not applicable.

  1. If the above patient is kept in the analysis the authors need to edit lines 321-322 “we enrolled only lung cancer patients undergoing pulmonary resection and excluded patients receiving ECMO support for other surgical indications, such as mediastinal and thymic resections.”

As we think that the Abruzzini case may add some value in a very small population study, we edit line 321 – 322 according your suggestion (Revised version new lines 334 – 335)

In conclusion, as previously mentioned this is an interesting study with clinical applicability in surgical practice. As such I am keen for it to be presented to the surgical community. However, there are some significant methodological and study conduction issues that need addressing prior to doing so.  I wish well to all.

My kindest regards,

Round 2

Reviewer 2 Report

Dear Editor and Authors,

I read and re-evaluated the revised manuscript as well as the responses given to my queries. The manuscript is much improved and I am satisfied to recommend its publication. Good job to all and congratulations.

Kind regards and hope to see you soon.

Emmanouil I. Kapetanakis, MD, MSc